# A Low-Cost Lightweight Deflectometer with an Arduino-Based Signal Interpretation Kit to Evaluate Soil Modulus

**DOI:** 10.3390/s23249710

**Published:** 2023-12-08

**Authors:** Huyen-Tram Nguyen, Yunje Lee, Jaehun Ahn, Taek Hee Han, Jun Kil Park

**Affiliations:** 1Department of Civil and Environmental Engineering, Pusan National University, Busan 46241, Republic of Korea; nhtram153@pusan.ac.kr (H.-T.N.); lee_yunje@pusan.ac.kr (Y.L.); 2Ocean Space Development & Energy Research Department, Korea Institute of Ocean Science and Technology, Busan 49111, Republic of Korea; taekheehan@kiost.ac.kr (T.H.H.); jkpark@kiost.ac.kr (J.K.P.)

**Keywords:** lightweight deflectometer, electro-mechanical, Arduino^®^, soil stiffness, soil modulus

## Abstract

This research presents an innovative solution aimed at addressing the cost and accessibility challenges associated with soil stiffness analysis in construction projects. Traditional lightweight deflectometer (LWD) systems have limitations due to their high cost and proprietary nature, prompting the need for a more widely accessible technology. To fulfill this purpose, a low-cost, open-source LWD onboard sensor signal interpretation system, utilizing Electro-Mechanical and Micro-Electro-Mechanical-System (MEMS) technology-based sensors in conjunction with an Arduino^®^ Uno and ADS1262 Breakout Board, has been developed. This system efficiently processes raw signal data into deflection and force units, enabling precise soil property analysis. Thorough enhancements, calibration, and alignment procedures have been applied and validated through field tests, which have produced highly satisfactory results. By significantly reducing costs while maintaining accuracy, this developed system has the potential to popularize quality control and assurance practices in the construction industry. This open-source approach not only enhances affordability but also broadens accessibility, making soil property analysis more efficient and attainable for a wider range of construction projects.

## 1. Introduction

The use of lightweight deflectometers (LWDs) in quality control and assurance testing for earthworks has increased in importance. They provide a rapid means of assessing the compaction quality or capacity of soils through reliable elastic modulus measurements [1,2,3]. One of the primary functions of LWDs is to detect the interaction between its mechanical components and the soil using sensors, where the force and deflection are induced during operation. The onboard sensors in LWDs are typically based on Electro-Mechanical and Micro-Electro-Mechanical System (MEMS) technologies, such as seismometers that measure acceleration or velocity and load cells that measure applied force.

Despite providing convenience in quality control, commercial LWD signal interpretation kits for LWDs often come with moderately high prices. The leading motivation behind this research is to reduce the production costs of LWDs, making this technology accessible to a wider audience. A production study of a cost-effective LWD sensor signal interpretation kit was carried out to fulfill this goal.

Soil compaction and bearing capacity can be quickly assessed from LWD measurement results by determining the relative relationships between soil properties, making LWDs a useful tool in construction and civil engineering. They provide data essential for ensuring the structural integrity of roadways, pavements, and other civil infrastructure. The use of LWDs is well documented in standards like ASTM E2835-11 [1]. The application of lightweight deflectometers (LWDs) has garnered significant attention in diverse research contexts. For instance, Marradi et al. [2] conducted extensive LWD tests on both actual roads and test tracks. Their work demonstrated the reliability of a performance-related approach for quality assurance/quality control (QA/QC) activities in roadworks. Another study presented in Kumar et al. [3] utilized an LWD to establish a target stiffness value for an unbound material. The compaction process was closely monitored until the material achieved the specified target stiffness. In Marecos et al. [4], researchers employed an LWD to determine the elastic modulus of the pavement subgrade. This approach facilitated the identification of damaged areas through the combination of various non-destructive methods. These studies collectively underscore the versatility and effectiveness of LWDs in diverse applications, ranging from QA/QC practices in road construction to targeted stiffness control in the subgrade soil and comprehensive assessment of pavement subgrade conditions.

In various fields of study, including in smart infrastructure, industry, medical applications, body control, aerospace, and laboratories, Arduino^®^ boards, like the Arduino^®^ Uno (ATmega328), have been widely adopted [5]. They offer a single-board microcontroller with programmable input/output peripherals, and they have been utilized in various monitoring applications due to their user-friendly operation, affordability, and moderate-to-high measurement accuracy [6,7,8]. The application of Arduino^®^ in geotechnical engineering is exemplified through the development of hardware and software solutions for estimating and regulating soil moisture, as demonstrated in the study of Garg et al. [9]. Zaman et al. [10] utilized geophone sensors and an Arduino^®^ microcontroller as the data acquisition center, which exemplifies an application of Arduino^®^ as a cost-effective approach to identify soil types and layers for preliminary geotechnical characterization. Sensors based on Electro-Mechanical and Micro-Electro-Mechanical-System (MEMS) technologies, along with external hardware components such as analog-to-digital converters and Arduino^®^ Uno boards, have the potential for use in various seismic and force signal monitoring devices [11]. For instance, Kafadar [12] developed a low-cost, computer-aided system that utilizes geophones to record, monitor, and analyze three-component microtremor data using an Arduino^®^ Uno. Ahmed et al. [13] investigated the use of MEMS accelerometers to measure vibration parameters with Arduino^®^ Uno as the data collection system. Itikala [14] created Arduino^®^ weighing equipment that uses a strain-gauge load cell to measure the weight produced by the load. Aravind et al. [15] utilized a load cell in conjunction with an Arduino^®^ Uno to measure fuel levels.

This study aims to address the implementation and application of a cost-effective, moderate-precision, controllable, and regularly calibrated LWD sensor signal interpretation kit by leveraging open-source hardware, specifically Arduino^®^ boards. The integration of Arduino^®^ boards with MEMS-based sensors and precision analog-to-digital converters (ADCs) is explored to enhance measurement accuracy. The limitations of Arduino^®^ boards, particularly in regard to dynamic behavior monitoring, are also discussed. In addition, the application of this signal interpretation kit is employed with an LWD to measure Young’s modulus of soils in the field, serving as a test of the usability and effectiveness of the newly developed model.

## 2. Implementation of an LWD

Figure 1 describes the components that make up the LWD and the sensors integrated inside it (ASTM E2835-11 [1]). Operation of the LWD begins by using the fix and release mechanism to drop the hammer, causing it to fall freely along the guide rod for a certain distance. After falling, the hammer impacts the rubber buffer unit, generating a load. This applied load is then transferred to the load cell through the buffer plate and is measured. The load acts on the plate, which rests on the soil. The seismic sensor integrated into the plate is used to measure the velocity or acceleration, and then the deflection is derived. The studied device includes two onboard sensors: a geophone to sense the velocity of soil movement and derive the displacement time history from this velocity, and a load cell to capture the applied force. The description of the mechanical frame and the basic principles of the sensors are also depicted in Figure 1 and discussed in detail in the following sections.

### 2.1. Mechanical Frame

Within the mechanical frame of the LWD (Figure 1), several integral components contribute to its functionality. Primarily, the fix and release part (2) acts as a mechanism to secure the hammer (3), which has a weight range of 10–20 kg, in a precise position, or this part initiates the hammer’s release into free fall. The hammer descends as a free-falling object, guided by the rod (4) with a typical length of 0.8 m. A rubber buffer unit (5), with a typical stiffness range of 200–700 kN/m, serves as a buffer during the hammer’s descent, ensuring the uninterrupted dynamic operation of the lightweight deflectometer (LWD). The buffer plate (6) fixes the buffers and transmits the applied force to the load cell. The housing (7) functions as a load transmission intermediate and safeguards the geophone. The circular plate (8), with a diameter in the range of 0.1–0.3 m, provides support for the entire LWD, ensuring equilibrium for the precise calculations of soil parameters. Additionally, the incorporation of bubble spirit level modules (1) aids users in achieving heightened measurement accuracy by ensuring the balanced placement of the LWD.

### 2.2. Sensors

#### 2.2.1. Geophone

A geophone is a ground motion velocity transducer. Its basic design includes a mass-spring system within a permanent magnetic field. When there is motion at the base, a spring-supported moving coil moves within this magnetic field, generating a voltage proportional to the velocity of the base at its ends. Its outstanding features include high sensitivity and the ability to respond to a wide range of frequencies, enabling it to detect various types of motion signals, even very small ones. Additionally, geophones are often compactly designed with sturdy casings, making them easy to integrate into various devices used in the fields of seismology and geophysics for measuring ground motion [16,17,18,19,20]. Figure 2 illustrates the design and some important specifications of the two types of geophones tested in this study.

The geophones function as mass-spring systems (Figure 1), representing a single-degree-of-freedom system governed by a second-order differential equation, which is expressed as follows:(1)$$m\frac{\partial^{2}x}{\partial t^{2}}+c\frac{\partial x}{\partial t}+kx=m\frac{\partial^{2}u}{\partial t^{2}}$$
where *m* is the proof mass (kg), *x* is the displacement of the proof mass relative to the sensor housing (m), *c* is the damping of the geophone inner spring (Ns/m), *k* is the stiffness of the geophone inner spring (N/m), and *u* is the ground or base displacement (m). Equation (1) can be rewritten as follows:(2)$$\frac{\partial^{2}x}{\partial t^{2}}+2\xi \omega_{0}\frac{\partial x}{\partial t}+\omega_{0}{}^{2}x=\frac{\partial^{2}u}{\partial t^{2}}$$
where $\xi =c/2\sqrt{km}$ is the open-circuit damping ratio, and $\omega_{0}=\sqrt{k/m}$ is the undamped natural frequency of the geophone (rad) [23,24].

The analog voltage from a geophone is equal to the product of its sensitivity and the velocity of the proof mass, which is expressed as follows:(3)$$V_{G-out}=S_{G}\frac{\partial x}{\partial t}$$
where $V_{G-out}$ is the output analog voltage induced by the geophone (V) and *S_G_* is the sensitivity constant of the geophone (V/m/s) [16].

From Equation (2), the displacement of the proof mass relative to case (*X*) in terms of the displacement of case (*U*) is shown in the frequency domain as follows:(4)$$-\omega^{2}X+2j\lambda \omega_{0}\omega X+\omega_{0}{}^{2}X=-\omega^{2}U$$
where ω is the signal frequency. It gives the following:(5)$$X=\frac{-\omega^{2}}{-\omega^{2}+2\zeta \omega_{0}\omega +\omega_{0}{}^{2}}U$$

An output signal is created through magnetic induction and is proportional to the proof mass velocity. In order to further express the role of the ground velocity in signal generation, Equation (5) needs to be written in terms of proof mass velocity and ground velocity as follows:(6)$$\frac{\partial X}{\partial t}=\frac{-\omega^{2}}{-\omega^{2}+2\zeta \omega_{0}\omega +\omega_{0}{}^{2}}\frac{\partial U}{\partial t}$$

By replacing the proof mass velocity with $V_{G-out}/S_{G}$, the equation for the geophone analog voltage in terms of ground motion can be achieved as follows:(7)$$V_{G-out}=-S_{G}\frac{\omega^{2}}{-\omega^{2}+2\zeta \omega_{0}\omega +\omega_{0}{}^{2}}\frac{\partial U}{\partial t}$$

Hence, the transfer function between ground velocity and output signal in terms of frequency can be defined as follows:(8)$$H_{G}=-S_{G}\frac{\omega^{2}}{-\omega^{2}+2\zeta \omega_{0}\omega +\omega_{0}{}^{2}}$$

The operational principle inside the sensor introduces some measurement errors. Figure 3 represents the normalized sensitivity and phase of the employed geophone as a function of frequency. Below the natural frequency, the output signal magnitude is attenuated. Correction is necessary to address these dynamic measurement errors in the frequency domain [25]. The correction of the dynamic measurement errors in the geophone signal will be discussed in Section 2.3.

#### 2.2.2. Load Cell

A load cell is a transducer that converts the force (weight) into a measurable electrical output. One of the widely utilized load cells is a four-arm Wheatstone bridge strain-gauge load cell (Figure 1), which is equipped with at least one precision strain gauge in each arm. The strain gauge is a sensor whose resistance value varies with changes in strain. A DC power voltage ($V_{ex}$ (V)) ranging from 5 to 20 V is applied to excite the bridges. Under no load conditions, all of the strain gauges have the same resistance value, resulting in a balanced bridge. When a load is applied to the load cell, the strain gauge deforms, creating an unbalanced bridge, generating the voltage ($V_{L-out}$ (V)) which is proportional to the applied load [26]. This voltage is expressed as a single differential voltage and estimated using the following equation [27]:(9)$$V_{L-out}=\left[\frac{R_{3}}{R_{3}+R_{4}}-\frac{R_{2}}{R_{1}+R_{2}}\right]V_{\mathrm{ex}}$$where *R*_1_, *R*_2_, *R*_3_, and *R*_4_ are the resistances (Ohms) of the strain gauges (Figure 1).

The output voltage is expressed in terms of a single differential voltage, which is then converted into a force value as follows:(10)$$Force=V_{L-out}\times \frac{R.C}{R\times V_{\mathrm{ex}}}$$
where R.C (rated capacity) is the maximum load that the load cell can bear and still operate under (>0.8 load) (tf), and *R* (rated output) is the output voltage that the load cell produces at the rated capacity per excitation volt at the input terminals minus the output voltage at the minimum load (mV/V). This study uses a low-profile, four-wire tension or compression load cell, model UL-T5, manufactured by Dacell [28], and the specifications of this load cell are shown in Table 1.

### 2.3. Signal Interpretation Kit and Processing

#### 2.3.1. Signal Interpretation Kit

The dynamic operation of the LWD leads to vibration of the system under impulsive loads within a specified frequency range from 0 to 500 Hz. This necessitates that the signal interpreter kit provides a sufficiently high sampling rate to avoid aliasing [29,30]. To prevent the loss of signal information for LWD dynamic signals with a bandwidth of *f*_0_, the sampling frequency *f_s_* must strictly exceed it to ensure perfect signal reconstruction from the samples. The *f_s_* is referred to as the Nyquist frequency [31].

To enhance the sampling rate of the interpreter system, a digital-to-analog (ADC) module, model ADS1262 (manufactured by TEXAS INTRUSMENT) [32], was integrated into the Arduino^®^ board. ADS1262 is a precision analog-to-digital converter (ADC) with a programmable gain amplifier (PGA) and voltage reference. The module can convert an analog signal from sensors with a high-resolution value of 32 bits, low-noise, and a data rate of up to 38,400 samples per second. The signal interpretation kit includes two Arduino^®^ Uno boards and two ADS1262 modules. Each sensor is connected to a separate Arduino^®^ board and ADS1262 module to prevent mutual interference during the measurement process. The connection diagram and each connection mode of the kit were created following the instructions provided by Venkatesh [33]. With this setup, both the geophone and the load cell were recorded at a speed of 2200 samples per second.

#### 2.3.2. Signal Processing and Calibration

##### Geophone Measurement

In the process of measuring the geophone signal, the analog-to-digital converter (ADS1262) [32] was responsible for receiving the analog voltage output from the geophone and subsequently converting it into a digital number. There is a linear correlation between the actual voltage value ($V_{G-out}$) and the voltage value obtained from the kit (*V_K_*). To evaluate the correlation, the voltage from the geophone was measured with the proposed kit (Arduino Uno and ADS1262) and compared to the voltage captured by a commercial oscilloscope (RTE 1034, Rohde and Schwarz [34]). This work was set up in-house, as shown in Figure 4. The vibration was induced in the geophone by applying a modest impact on the table. The first peak value of the output voltage captured by the oscilloscope (*V*_0_) was used as the reference value of the voltage generated by the geophone ($V_{G-out}=V_{0}$) to calculate the ratio between the two voltage signals (Table 2). The mean of *V*_0_/*V_K_* was used as the correlation to evaluate the geophone signal.

The signal interpretation kit causes noise, and this noise depends on the data rate and PGA gain of the kit. The work of Szymanowski [35] implemented an exponentially weighted least squares (EWLS) algorithm to detect and cancel the impulsive noise in recorded music using MATLAB R2022b software [36]. This method was applied to eliminate the impulsive noise induced by the signal interpretation kit for both the geophone and load cell signal (the load cell signal is implemented after calibration in the next section). Figure 5 displays the signal after filtering out the noise.

After obtaining the enhanced and removed noise output voltage (*V_A_*) time history, the signal is corrected due to the dynamic measurement error mentioned in Section 2.2.1. The correction process begins with the Fast Fourier Transform (FFT) of the sensor’s output voltage after filtering and calibrating, which is recorded in the time domain. The FFT transforms the signal to be in the frequency domain. Next, the transformed signal is multiplied by the inverse transfer function (Equation (8)). Finally, the result is brought back to the time domain by applying an Inverse Fast Fourier Transform (IFFT). These steps effectively eliminate the dynamic measurement error introduced by the geophone [23], as shown in Figure 6. This process is carried out based on the code provided by Stamp [23].

The maximum displacement is then derived by integrating the velocity with time. This study performs numerical integration via the trapezoidal method following the suggestion of Stamp and Mooney [25]. The trapezoidal method approximates the integration over an interval by breaking the area down into trapezoids with more easily computable spaces, according to the following equations [37]:(11)$$x(t)=\int_{a}^{b}v(t)dt\approx \frac{b-a}{2N}\sum_{n=1}^{N}\left[v(t_{n})+v(t_{n+1})\right]$$
where *x*(*t*) is the derived displacement (m), and *v*(*t*) is the velocity (m/s). While converting from velocity to displacement, numerical integration errors can occur. These errors are inherent in the numerical approximation process due to the discrete nature of the data and the chosen approximation method. Fortunately, such errors can be mitigated by employing a polynomial linear baseline correction methodology on the raw derived displacement signal. An example of a displacement time history, derived from both corrected and uncorrected velocity data after removing the numerical integration errors, is illustrated in Figure 7.

##### Load Cell Measurement

Calibration of the load cell is performed to ensure measurement accuracy. Initially, the digital reading is recorded when the load cell is in an unloaded state. Subsequently, objects with known physical weight values (within the range of 1–25 kg) are selected. These objects are placed on the load cell one by one, and the output digital reading for each object is recorded sequentially. The list of calibration factors is calculated as follows [38]:(12)$$C_{i}=\frac{D_{i}-D_{0}}{W_{i}}$$
where *D_i_* is the load cell measurement when the *i*th weight is applied, *D*_0_ is the load cell measurement with a zero load, and *W_i_* is the *i*th weight (reference). The mean *C* of the *C_i_* values is then evaluated to calibrate the measured weight. Following that step, the measurement value of the load cell is calibrated as follows:(13)$$F_{1}=\frac{\left(D_{m}-D_{0}\right)}{C}$$
where *F*_1_ is the calibrated load cell measurement, and *D_m_* is the digital reading of the load. Figure 8 shows this comparison of the measured and calibrated loads and reference weights.

The overall process for carrying out sensor signal calibrations and interpretations is summarized in Figure 9.

## 3. Materials and Testing Results

### 3.1. Materials

For the test, two types of open-grade aggregates (OGAs) were selected for the measurements. These materials are identical to those used in the previous study by Choi et al. [39]. The maximum particle sizes of these aggregates are 40 mm (D40) and 25 mm (D25), and they are both rhyolite rocks. D40 possesses a coefficient of uniformity of 2.88 and a coefficient of curvature of 1.19, while D25′s coefficients are 2.48 and 1.02, respectively. These materials fall within a specific gravity range of 2.67 to 2.75 [39].

The tests were conducted on the Yangsan campus of Pusan National University in South Korea (Figure 10). The area was excavated, filled, and compacted, with a material thickness of 0.6 m [39].

### 3.2. Testing Results

From the derived deflection and applied force of the sensors of the low-cost kit, the modulus ($E_{LWD}$) is calculated based on the currently used Boussinesq solution as follows [23]:(14)$$E_{LWD}=\frac{A(1-\nu^{2})F_{pk}}{\pi rd_{pk}}$$
where *A* is the stress distribution factor (being 2 in this study), *v* is Poisson’s ratio (being 0.35 in this study), *F_pk_* = the peak force applied on the plate (N), *r* is the radius of the plate (m), and $d_{pk}$ is the peak displacement at the center of the plate (m).

In addition, the Zorn LWD [40] does not have a load cell and assumes a peak applied load of 7.07 (kN) at the full falling height value of 72.4 (cm), regardless of the stiffness of the soil. An alternative to the load cell is to use theory to calculate the applied force, as in Equation (15), with a spring stiff constant *K* = 362,396 (N/m). The theory of applied force neglects the discrepancies between theoretical and experimental applied forces because of the non-linear buffer behavior during loading [41,42].
(15)$$F=\sqrt{2\times g\times m\times h\times K}$$
where *F* is the applied force (N), *m* is the mass of the hammer (kg), *h* is the falling height of the hammer (m), and *g* is the acceleration of gravity, i.e., 9.81 (m/s^2^).

Table 3 presents our testing results from two different sites with varying values of the falling height of the hammer. Onboard sensors were utilized to measure the applied force and plate movement velocity to derive displacement. These two values were subsequently analyzed to calculate the elastic modulus using Equation (14).

The results indicate that with a larger falling height of the hammer, both the applied force and deflection are greater, so that the differences in falling height of the hammer result in negligible variation in the elastic modulus evaluation. Additionally, the results of the measured force and the derived deflection show very good agreement between different trial measurements, leading to negligible differences in the estimated elastic modulus. The results of the newly developed LWD were systematically compared with those of Choi et al. [43], who had extensively used the ZFG 3000 (Zorn Instruments, Stendal, Germany [40]) and Soil Stiffness Gauge (SSG, Humboldt, IL, USA [44]) to determine the modulus of elasticity of D40 and D25 materials. Their measurements showed that for the D40 material with a dry density from 15.39 to 16.25 kN/m³, the ZFG 3000 gave values between 25.50 and 42.10, while the SSG measurements ranged from 38.20 to 49.18 kN/m³. Similarly, for the D25 material with a dry density varying from 14.99 to 16.07 kN/m³, the ZFG 3000 gave values between 27.41 and 41.77, and the SSG measurements ranged from 38.29 to 47.30 kN/m³. The comparative analysis showed that the results from the newly developed LWD model showed minimal deviations, falling within a narrow range when compared to the measurement techniques commonly used today. The measurement results are stable and within the expected elastic modulus range for the materials, demonstrating the potential usability of the model introduced in this article.

In comparing the experimental results obtained for the measured force and the assumption force, the calculated theoretical force is approximately 38% greater than the measured value. This discrepancy also results in a similar variation in the calculated elastic modulus outcomes.

Davich [45] observed that the use of theoretical force (no force measurement) can lead to an overestimation of the modulus by 4% to 8% for soft to very stiff soils. The test results of Davich et al. [46] based on Keros Prima [47] also indicate about an 8% overestimation of the elastic modulus when the theoretical force was assumed. The observed variation in measurement outcomes resulting from the utilization of both measured and assumed forces, as well as the inconsistencies in commercial LWD measurements, emphasizes the need for future research that is focused on rectifying these inconsistencies and crafting a more reliable LWD [25,42].

## 4. Conclusions

In this study, a low-cost signal interpretation kit was developed for analyzing signals from the sensors in a lightweight deflectometer (LWD) system. Notable features of this developed signal interpretation kit include its affordability (approximately USD 180), open-source nature, ease of replication, and user-friendliness.

A sensor signal interpreter underwent validation for signal interpretation. It involved enhancing measurement signals by employing noise mitigation principles known as the EWLS algorithm. Simultaneously, signals from the geophone were calibrated using the oscilloscope, a standard measuring device, while signals from the load cell were calibrated using reference weights. Then, a frequency domain signal correction was implemented to the geophone signal. The affordability of this kit broadens accessibility to LWD measurements, allowing a diverse user base without the need for extensive experience and resources typically associated with expensive commercial products.

The field test results indicate that a larger failing height of the hammer leads to a larger applied force and deflection, which are compensated for during the modulus evaluation, resulting in negligible changes in the elastic modulus. The measured force and derived deflection show good agreement across trials, with stable results within the expected elastic modulus range. When comparing the measured and theoretical force, the theoretical force was higher for most cases, causing discrepancy in elastic modulus evaluations, which corresponds with the qualitative findings reported in the literature.

Further investigations are required to identify the source of inconsistency in general lightweight deflectometer (LWD) devices, which may be due to their mechanical structure and integrated sensors, as elucidated in the literature and this study. This will be achieved through a comprehensive comparative analysis, employing alternative methods for estimating the elastic modulus. In addition, several aspects of the LWD are intended to be adjusted to provide more accurate and consistent results. One of the important aspects is to enhance the dynamic measurement calibration process of the sensors, for instance, using an accelerometer during load cell calibration to account for dynamics regarding impact application and force measurement.

## Figures and Tables

**Figure 1 sensors-23-09710-f001:**
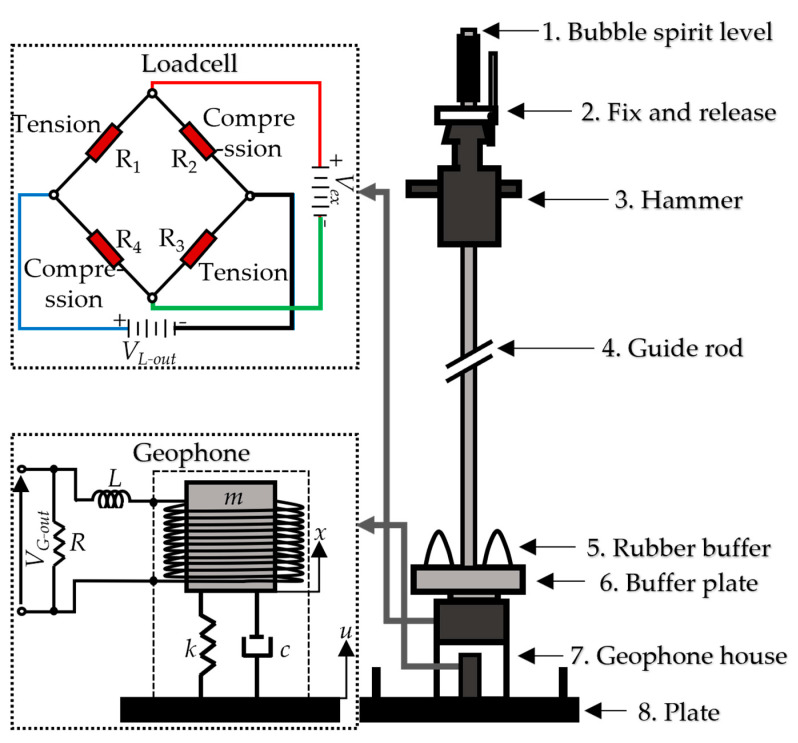
Schematic of LWD mechanical frame, onboard sensors, and circuits.

**Figure 2 sensors-23-09710-f002:**
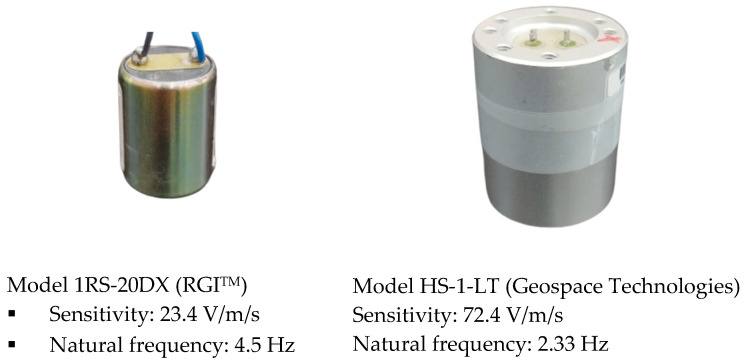
Two types of different natural frequency geophones: one with a higher natural frequency (**left**) and the other with a lower natural frequency (**right**) [21,22].

**Figure 3 sensors-23-09710-f003:**
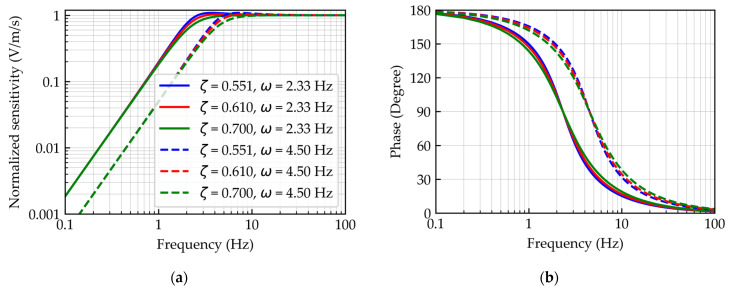
Geophone normalized sensitivity and transfer function phase with different values due to open-circuit damping and natural frequency. (**a**) Normalized sensitivity; (**b**) transfer function phase.

**Figure 4 sensors-23-09710-f004:**
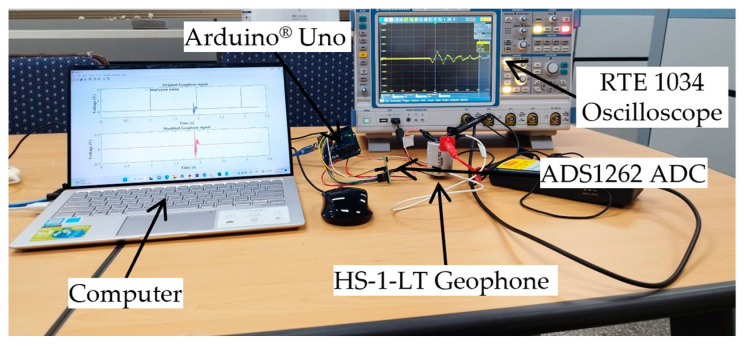
Test set up to evaluate the mutation parameter of the signal interpretation kit.

**Figure 5 sensors-23-09710-f005:**
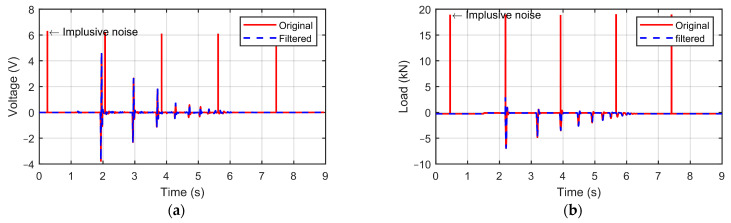
Impulsive noise removal implemented in MATLAB R2022b. (**a**) Original and filtered geophone signals. (**b**) Original and filtered load cell signals.

**Figure 6 sensors-23-09710-f006:**
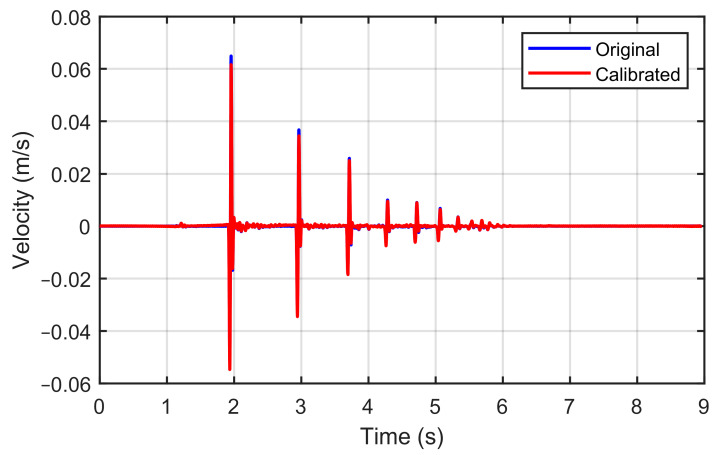
Geophone’s velocity signal with and without dynamic measurement correction.

**Figure 7 sensors-23-09710-f007:**
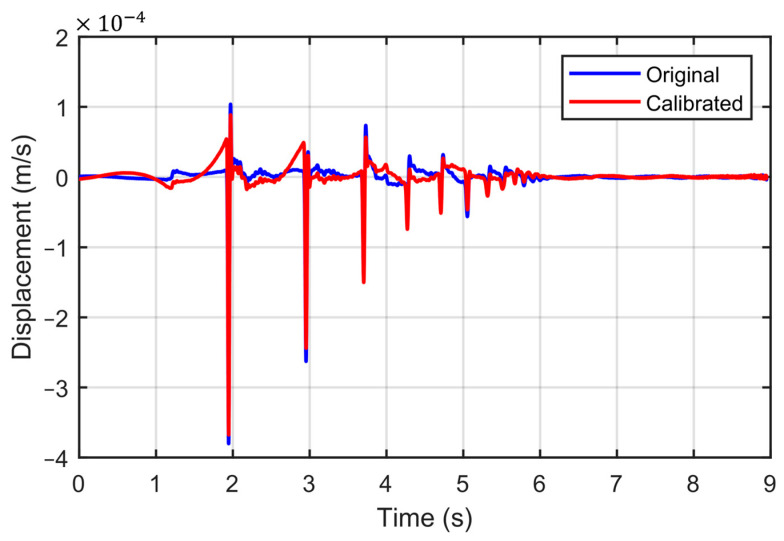
Displacement history derived from the velocity history.

**Figure 8 sensors-23-09710-f008:**
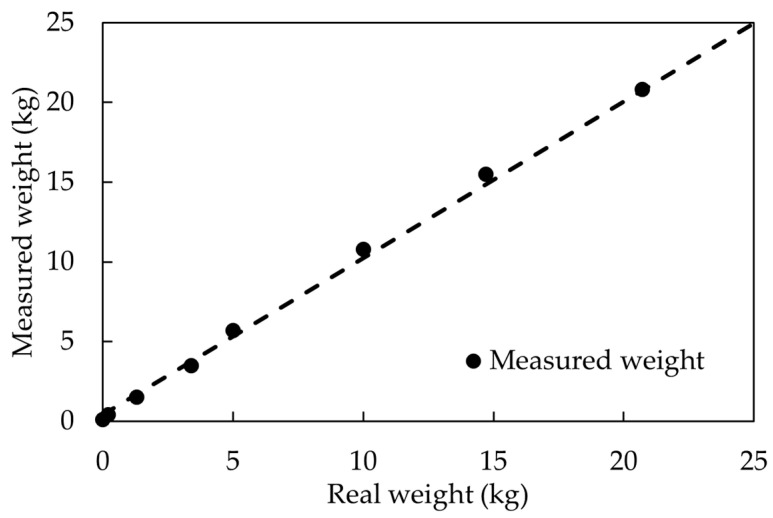
Load cell measurement comparison.

**Figure 9 sensors-23-09710-f009:**
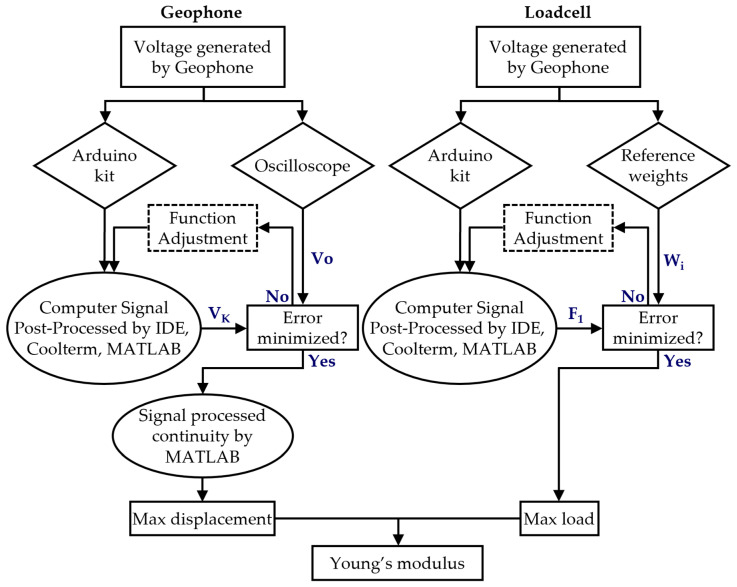
Diagram describing LWD’s signal interpretation and calibration process.

**Figure 10 sensors-23-09710-f010:**
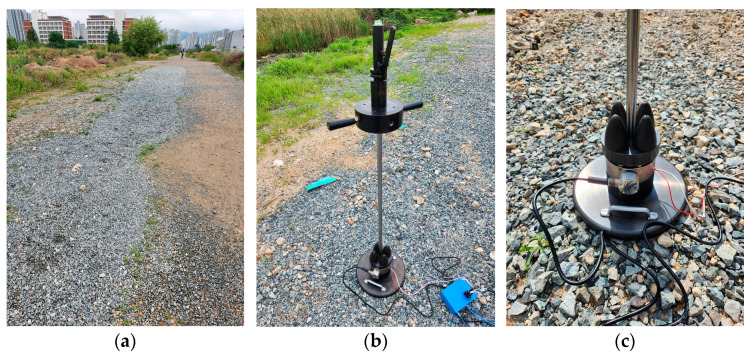
Lightweight deflectometer (LWD) testing in the field. (**a**) LWD testing sites in the field; (**b**) LWD equipment at the site; (**c**) zoomed-in view of LWD.

**Table 1 sensors-23-09710-t001:** Specification of the load cell used in this study [28].

Rated Capacity (tf)	Rated Output (mV/V)	Excitation Voltage (V)
5	2 ± 0.1%	5~10

**Table 2 sensors-23-09710-t002:** Calibration of geophone signal.

No.	Oscilloscope Voltage: *V*_0_ (mV)	*V_K_* (mV)	*V*_0_/*V_K_*
1	2120	1112.10	1.91
2	2403	1057.50	2.27
3	2603	1167.40	2.23
4	2606	1213.80	2.15
5	2080	952.20	2.18
6	3436	1487.20	2.31
7	3930	1800.90	2.18
Mean = 2.18			

**Table 3 sensors-23-09710-t003:** Test results.

Material	Falling Height (m)	Measurement Force ^1^ (*F*_1_) (kN)	Calculation Force (*F*_2_) (kN)	$E_{LWD1}$ from *F*_1_ (MPa)	$E_{LWD2}\;\mathrm{from}\;F_{2}$ (MPa)
D40	0.66	4678	7548	58.82	94.95
	0.76	5010	8099	59.30	95.88
D25	0.66	4464	7548	47.30	79.98
	0.76	4781	8099	46.77	79.23

^1^ Three measurements for each option.

## Data Availability

Data are contained within the article.

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
