# Peer review of "A Low-Cost Lightweight Deflectometer with an Arduino-Based Signal Interpretation Kit to Evaluate Soil Modulus"

_sensors, 2023, doi:10.3390/s23249710_

Round 1

Reviewer 1 Report

Comments and Suggestions for Authors

In abstract line 18, do not use word I, our, we and us. rephrase sentence.

Line 25-28 please check how soil elastic modulus can measure by deflectometer. What is the use oedometer or triaxial test?

Similarly, line 39 check your statement of LWD measures compaction and bearing capacity.

Line 77 “This study aims at addressing the implementation and application of a cost-effective

LWD sensor signal interpretation kit by leveraging open-source hardware, specifically Arduino® boards.” This is the novelty of paper then it is not suitable for Impact factor Journal. Bring it something more to make it worthy for good Journal such higher efficiency of measurement, quick etc.

How Authors chose critical damping (damping ratios) for the measurement? The values seem high for real structures interacted with soil.

There is no cross checking of results/data gathered through newly developed sensor with well-established technique. Please add.

Comments on the Quality of English Language

Language of the manuscript needs improvement

Author Response

COVER LETTER OF MANUSCRIPT ID Sensors-26888115

Dear Reviewers and Editors,

First of all, I would like to thank all of you for taking time to read my manuscript and sending me your insight comments and suggestions. Your insights have helped us to improve the quality of this paper. I have generally edited and checked some parts of the paper as follows:

  1. The title of the paper was changed to “A Low-Cost Lightweight Deflectometer with an Arduino-Based Signal Interpretation Kit to Evaluate Soil Modulus”
  2. I have checked all references relevant to the content of the manuscript and there are no problems.
  3. I have highlighted in yellow the revision of the manuscript due to the reviewers' comments. And I have highlighted in blue the revision of the manuscript due to the need for revision.
  4. The English writing throughout the manuscript has been extensively grammatically and semantically revised using the MDPI editing service (ID english-73852) and is certified as an attached certificate. In the debug version of the MDPI editing service, there are many small errors in the English writing that have been corrected without semantic changes, so I have not highlighted these parts.
  5. There are some revisions related to the figures and table of the paper as follows:
    • Figure 1. is completely drawn by the authors. The figure generally describes the Lightweight Deflectometer (LWD) as mentioned in ASTM E2835-11. (The citation of ASTM E2835-11 has been added in line 89 of the old manuscript and in line 90 of the revised manuscript).
    • Figure 2. is owned by the authors. The specification of products specified in references 24,25 of the old manuscript and references 21,22 of the revised manuscript.
    • Figure 4 was deleted because of its unnecessary appearance. The text is also modified due to the deletion of the figure (delete “as depicted in Figure 4” in line 192 of the old manuscript).
    • Figure 5 and Table 2 were deleted because of their unnecessary appearance. All the information can be found in Reference 33. The text is also modified due to the deletion of the figure and table (replace “depicted in Figure 5. And Table 2 shows each connection mode of the kit in detail.” by “The connection diagram and each connection mode of the kit were created following the instructions provided by Venkatesh [33].” in line 208 of the old manuscript and 203-204 of the revised manuscript).
    • Figure 11. The idea and drawing are created by the manuscript authors.
    • Figures 12, 13, and Table 4. were deleted because of their unnecessary appearance. All the information can be found in Reference 39. The text is also modified due to the deletion of the figure and table (replace “Figure 13 shows the lower and upper bounds of the particle size distribution of two typical aggregate base layers above the subbase: ASTM No. 57 and AASHTO M43 [44,45], and compares them to the particle size distribution of the measured materials.” by “The maximum particle sizes of these aggregates were 40 mm (D40) and 25 mm (D25), and they are both rhyolite rocks. D40 possesses a coefficient of uniformity of 2.88 and a coefficient of curvature of 1.19, while D25’s coefficients are 2.48 and 1.02, respectively. These materials fall within a specific gravity range of 2.67 to 2.75 [39].” In line 289 of the old manuscript and in line 284-287 of the revised manuscript).

The following are my point by point and the details of the revisions and my responses to the referees’ comments and questions.

RESPONSE TO REVIEW REPORT FORM 1

Comments and Suggestions for Authors

Your comment_1: “In abstract line 18, do not use word I, our, we and us. rephrase sentence.”

Discussion: I have changed the sentence structure and removed the word "our" in abstract line 18 (old manuscript) and in abstract line 20 (revised manuscript) without changing the meaning of the sentence as follows: “By significantly reducing costs while maintaining accuracy, our system has the potential to popularize quality control and assurance practices in the construction industry.”-> “By significantly reducing costs while maintaining accuracy, this developed system has the potential to popularize quality control and assurance practices in the construction industry.”

Your comment_2: “Line 25-28 please check how soil elastic modulus can measure by deflectometer. What is the use oedometer or triaxial test?”

Discussion: I would like to explain more detail as follows

  1. I have checked and rewritten the sentences lines 25-28 (old manuscript) and in lines 27-32 (revised manuscript) to explain more detail about how soil elastic modulus can be measured by lightweight deflectometers as follows: “Lightweight deflectometers (LWDs) have gained increasing importance in quality control and assurance testing for earthwork. They provide a rapid means of assessing the compaction quality or capacity of soils through reliable elastic modulus measurements [1-3]. One of the primary functions of the LWD is to use sensors to detect the interaction between its mechanical components and the soil, where the force and deflection are induced during the operation.” -> “The use of lightweight deflectometers (LWDs) in quality control and assurance testing for earthworks has increased in importance. They provide a rapid means of assessing the compaction quality or capacity of soils through reliable elastic modulus measurements [1-3]. One of the primary functions of LWDs is to detect the interaction between its mechanical components and the soil using sensors, where the force and deflection are induced during operation.”
  2. In this paper, we do not study and deploy the oedometer or triaxial test.

Your comment_3: “Similarly, line 39 check your statement of LWD measures compaction and bearing capacity.”

Discussion: To further clarify the statement of LWD measures compaction and bearing capacity, I would like to explain in more detail as follows:

  1. The Lightweight Deflectometer is used to quickly measure the modulus of elasticity of soils, from which compaction and bearing capacity can be assessed.
  2. The sentence has been rewritten to clarify the statement "LWD measures compaction and bearing capacity" in line 39 (old manuscript) and in lines 40-42 (revised manuscript) without changing the meaning of the sentence as follows: “LWDs have become crucial tools in construction and civil engineering due to their ability to assess soil compaction and bearing capacity quickly and accurately.”-> “Soil compaction and bearing capacity can be quickly assessed from LWD measurement results by determining the relative relationships between soil properties, making LWDs a useful tool in construction and civil engineering.”

Your comment_4: “Line 77 “This study aims at addressing the implementation and application of a cost-effective LWD sensor signal interpretation kit by leveraging open-source hardware, specifically Arduino® boards.” This is the novelty of paper then it is not suitable for Impact factor Journal. Bring it something more to make it worthy for good Journal such higher efficiency of measurement, quick etc.”

Discussion: First of all, I would like to thank you for your comments on the novelty of our paper. In order to clarify the research objectives of the paper, I have explained in more detail as follows:

  1. In this paper we use open-source hardware, specifically Arduino® boards, to create our own LWD. The most outstanding feature of open-source hardware Arduino® boards, which has made them widely used in many fields to date, is their low cost, which allows them to reach a wide range of users who do not have the resources for expensive commercial products. This is why I have included it as a new point in the paper. In addition to this feature, it also has a number of other features such as: moderate accuracy, ease of use, which were not mentioned in the manuscript.
  2. The sentence has been rewritten to add the novelty of the paper that was previously missing before in line 77 (old manuscript) and in line 79-81 (revised manuscript) as follows: “This study aims at addressing the implementation and application of a cost-effective LWD sensor signal interpretation kit by leveraging open-source hardware, specifically Arduino®”-> “This study aims to address the implementation and application of a cost-effective, moderate-precision, controllable, and regularly calibrated LWD sensor signal interpretation kit by leveraging open-source hardware, specifically Arduino® boards.”

Your comment_5: “How Authors chose critical damping (damping ratios) for the measurement? The values seem high for real structures interacted with soil.”

Discussion: The critical damping (damping ratios) introduced in the paper is geophone relative to ground velocity critical damping. Its value is provided by Geophone manufacturer (Geo Space Technologies, Inc. Houston, Tx)

Your comment_6: “There is no cross checking of results/data gathered through newly developed sensor with well-established technique. Please add”

Discussion: There is no cross checking of results/data gathered through newly developed sensor with well-established technique. I have added measurement results from literature studies on the same material that I have studied on:

  1. A result from the same well-established technique of the commercial Zorn LWD.
  2. Another result from a different method of measurement is the SSG (Soil Stiffness Gauge).

The comparative sentences have been added to the revised manuscript on lines 319-329 (revised manuscript) as follows: “The results of the newly developed LWD were systematically compared with those of Choi et al. [43], who had extensively used the Zorn LWD (ZFG 3000 GPS, Zorn Instruments [40]) and SSG tests [44] to determine the modulus of elasticity of D40 and D25 materials. Choi, Ahn, Lee, and Ahn's [43] measurements showed that for the D40 material with a dry density from 15.39 to 16.25 kN/m³, the Zorn LWD gave values between 25.50 and 42.10, while the SSG measurements ranged from 38.20 to 49.18 kN/m³. Similarly, for the D25 material with a dry density varying from 14.99 to 16.07 kN/m³, the Zorn LWD gave values between 27.41 and 41.77 and the SSG measurements ranged from 38.29 to 47.30 kN/m³. The comparative analysis showed that the results from the newly developed LWD model showed minimal deviations, falling within a narrow range when compared to the measurement techniques commonly used today.”

Comments on the Quality of English Language

Your comment: “Language of the manuscript needs improvement.”

Discussion: The manuscript was carefully revised grammatically and semantically by using MDPI English editing service.

RESPONSE TO REVIEW REPORT FORM 2

Comments and Suggestions for Authors

Your comment: “The manuscript titled "A Low-Cost Lightweight Deflectometer with Arduino-Based Signal Interpretation Kit to Evaluate Soil Modulus" presents an intriguing contribution to the field of engineering. It focuses on the development of an open-source sensor kit designed to reduce the costs associated with proprietary tools used in evaluating soil modulus. This review aims to provide an overview of the manuscript's strengths and areas that require improvement.”

Discussion: Thank you very much for your comments, suggestions, and overview of the manuscript's strengths and for pointing out the improvement part of our manuscript. Your comments helped us to see our weaknesses and to improve the paper.

Your comment: The manuscript is well-structured, providing a clear outline of the problem, methodology, results, and discussions. The introduction effectively introduces the problem statement and the significance of the research. The "Implementation of LWD" section is detailed, making it easy for readers to understand the technical aspects of the study. The results and discussions are logically organized and complemented with figures and tables, enhancing the overall readability of the manuscript. However, it's worth noting that the text in Figure 5 (schematic) is too small for reading and should be addressed.”

Discussion: Thank you very much for your comment on the structure of our paper. Regarding the issue of figure 5. I have considered its necessity and realized that it can be replaced with reference documents, so I have removed it and explained the schematic with reference documents as in the revised manuscript lines 203-205 as follows: “The connection diagram and each connection mode of the kit were created following the instructions provided by Venkatesh [33].”

Your comment: While the manuscript's overall organization is strong, there are some sentences that are challenging to understand. These issues may stem from grammatical errors, awkward phrasing, or a lack of clarity in conveying the intended message. For instance, sentences like "One of the primary functions of the LWD is to use sensors to detect the interaction between its mechanical components and the soil, where the force and deflection are induced during the operation" could be made more straightforward. Careful proofreading and editing would significantly improve the manuscript's readability.”

Discussion: Thank you very much for your comment on the overall organization and English writing of our paper. We acknowledge the shortcomings in the sentences and the English writing problems throughout the paper. We have reviewed and rewritten the sentences to make them more understandable and straightforward. For example, in lines 25-28 (old manuscript) and in lines 30-32 (revised manuscript): "One of the primary functions of the LWD is to use sensors to detect the interaction between its mechanical components and the soil where the force and deflection are induced during operation." -> " One of the primary functions of LWDs is to detect the interaction between its mechanical components and the soil using sensors, where the force and deflection are induced during operation.”

Your comment: “The manuscript addresses a relevant and practical issue in geotechnical engineering by attempting to reduce the cost of evaluating soil modulus. The use of an Arduino-based signal interpretation kit to achieve this goal is innovative and aligns with the increasing trend of open-source solutions in science and engineering. However, there are areas where the manuscript could benefit from additional detail to fully convey the novelty of the approach.”

Discussion: I would like to thank you for your comments on the novelty of our paper and to clarify the novelty of the approach, some additional details are added in line 77 (old manuscript) and in lines 79-81 (revised manuscript) as follows: “This study aims at addressing the implementation and application of a cost-effective LWD sensor signal interpretation kit by leveraging open-source hardware, specifically Arduino® boards.”-> “This study aims to address the implementation and application of a cost-effective, moderate-precision, controllable, and regularly calibrated LWD sensor signal interpretation kit by leveraging open-source hardware, specifically Arduino® boards.”

Your question_1: “Why did you choose to use a dual microprocessor setup? The ADS1262 has an SPI (or similar) interface with a CS pin that could accommodate multiple units on a single Arduino. Alternatively, all 8 inputs on a single ADS unit could be utilized.”

Discussion: To more clarify the reason of choosing a dual microprocessor setup, I would like to explain in more detail as follows:

  1. To simplify the connection of the circuits, data processing coding and to avoid collisions that could lead to errors during the measurement process of the lightweight deflectometer (LWDs).
  2. The ADS1262 is used to increase the sampling rate of the kit. I have tried to read signals from two sensors with a single Arduino and ADS1262, but the sampling rate was unstable. To stabilize the sampling rate and ensure smooth signal processing, I decided to use a dual microprocessor setup.

Your question _2: “How did you synchronize both microcontrollers? The observed error could potentially result from this lack of synchronization.”

Discussion: I would like to explain how I have synchronized the two microcontrollers and why I have neglected the possibility of a result error due to this lack of synchronization: The main operation of the LWD is to create a light impact from which its on-board sensors record the history of that impact in terms of impact force and soil velocity. The aim is to record the peak value of these two physical quantities in order to calculate the modulus of elasticity of the soil. These peaks occur almost simultaneously (the force transmitted down the plate causes deflection). The dual microprocessor is wired and set up to operate in a similar way and has the same sampling frequency (2200Hz). The dual microprocessor operates throughout the LWD measurement process, the maximum values measured by the dual microprocessor are almost simultaneous (there is a very small difference due to the difference in the time of stress transmission to the plate). For this reason, I had not considered the possibility of an error in the results due to this lack of synchronization.

Your question _3: “You are comparing a static load on a load cell and a dynamic load. As the stress wave needs time to travel through the load cell and strain gages, the dynamic characteristics may not always match the static ones. Higher strain rates generally result in signal attenuation. Could this be the potential cause of errors? Testing the load cell's dynamic characteristics by striking it with a weight attached to an accelerometer (F = m*a) could provide insights.”

Discussion: First of all, I would like to thank you for your comment on the subject of load cell calibration. It is very helpful in shedding more light on what I want to present about the situation and the process of how I calibrate the loadcell.

It is true that there is a difference in load cell calibration when the force measured is a static load on a load cell and a dynamic load. As the stress wave takes time to travel through the load cell and strain gages, the dynamic characteristics may not always match the static ones.

 The strain gauge load cell used in this paper is a highly accurate and intensive load cell, capable of capturing rapid and real-time data, which is essential for dynamic measurements. They can capture transient changes in load, making them suitable for applications where load profiles change rapidly. Where their response frequency can reach up to 5 thousand Hertz, it helps loadcell measure real-time data during the changes in applied load. In addition, during operation, when the hammer falls and collides with the buffer (conical rubber buffer), there is a delay to allow the stress wave to propagate through the buffer to the loadcell. This delay gives the loadcell enough time to settle and give accurate readings with low error. This is why I did not consider calibrating the loadcell with a dynamic load.

Your method striking loadcell with a weight attached to an accelerometer (F = m*a) is a good solution for calibrating loadcells with dynamic loads. We would like to thank you for your insightful suggestions. We have included your suggestion as an important aspect of the future plan for further research in the conclusion section. The added sentences to the revised manuscript on lines 370-373 (revised manuscript) are as follows: “In addition, several aspects of the LWD are intended to be adjusted to provide more accurate and consistent results. One of the important aspects is to enhance the dynamic measurement calibration process of the sensors, for instance using an accelerometer during loadcell calibration to account for dynamics regarding impact application and force measurement.”

Your question _4: “Can you describe the method for obtaining displacement from velocities? What are your assumptions about initial conditions?”

Discussion: I would like to explain how I obtain displacements from velocities: I have derived the displacement time history from the velocity time history to data in MATLAB, the method I have used is trapezoidal numerical integration. The input units are time with an interval of 0.00045 and the velocity time history recorded by the developed kit. The initial velocity is the velocity at t=0, v (0)=0.

Your comment: “The manuscript sometimes delves into well-known facts, which can be redundant for readers familiar with the field. It would be more advantageous to use this space to provide further context, additional examples, or practical insights, which would enhance the manuscript's value.”

Discussion: After listening to your comments. I noticed that there are some well-known facts that are generally known, so I left out some well-known facts and added some additional examples as follows:

  1. Widely use commercial ZFG 3000 GPS from Zorn Instruments.
  2. Well-known elastic modulus determination test SSG (the soil stiffness gauge).

The added sentences to the revised manuscript on lines 319-329 (revised manuscript) as follows: “The results of the newly developed LWD were systematically compared with those of Choi et al. [43], who had extensively used the Zorn LWD (ZFG 3000 GPS, Zorn Instruments [40]) and SSG tests [44] to determine the modulus of elasticity of D40 and D25 materials. Choi, Ahn, Lee, and Ahn's [43] measurements showed that for the D40 material with a dry density from 15.39 to 16.25 kN/m³, the Zorn LWD gave values between 25.50 and 42.10, while the SSG measurements ranged from 38.20 to 49.18 kN/m³. Similarly, for the D25 material with a dry density varying from 14.99 to 16.07 kN/m³, the Zorn LWD gave values between 27.41 and 41.77 and the SSG measurements ranged from 38.29 to 47.30 kN/m³. The comparative analysis showed that the results from the newly developed LWD model showed minimal deviations, falling within a narrow range when compared to the measurement techniques commonly used today.” By adding some examples from previous studies to compare with the results of the new model. I hope this will make my manuscript more valuable.

Your comment: “In summary, "A Low-Cost Lightweight Deflectometer with Arduino-Based Signal Interpretation Kit to Evaluate Soil Modulus" is an intriguing manuscript with a solid foundation. With some improvements in language and a more thorough exploration of the novelty and applications of the open-source sensor kit, this research has the potential to make a more significant contribution to the field of geotechnical engineering.”

Discussion: Thank you very much for your comments on our paper.

Comments on the Quality of English Language

Your comment: “While the manuscript's overall organization is strong, there are some sentences that are challenging to understand. These issues may stem from grammatical errors, awkward phrasing, or a lack of clarity in conveying the intended message. For instance, sentences like "One of the primary functions of the LWD is to use sensors to detect the interaction between its mechanical components and the soil, where the force and deflection are induced during the operation" could be made more straightforward.”

Discussion: The manuscript was carefully revised grammatically and semantically by using MDPI English editing service. I have edited the instance sentence that you gave me as follows: “One of the primary functions of the LWDs is to use sensors to detect the interaction between its mechanical components and the soil using sensors, where the force and deflection are induced during the operation.” In line 25-28 of the old manuscript and in line 30-32 of revised manuscript.

Your suggestions and comments have helped us a lot in the process of editing and completing this paper. Once again, we would like to thank you. We hope that you will accept our statement.

Sincerely thank you.

Revised version submission date

14 November 2023

Reviewer 2 Report

Comments and Suggestions for Authors

The manuscript titled "A Low-Cost Lightweight Deflectometer with Arduino-Based Signal Interpretation Kit to Evaluate Soil Modulus" presents an intriguing contribution to the field of engineering. It focuses on the development of an open-source sensor kit designed to reduce the costs associated with proprietary tools used in evaluating soil modulus. This review aims to provide an overview of the manuscript's strengths and areas that require improvement.

The manuscript is well-structured, providing a clear outline of the problem, methodology, results, and discussions. The introduction effectively introduces the problem statement and the significance of the research. The "Implementation of LWD" section is detailed, making it easy for readers to understand the technical aspects of the study. The results and discussions are logically organized and complemented with figures and tables, enhancing the overall readability of the manuscript. However, it's worth noting that the text in Figure 5 (schematic) is too small for reading and should be addressed.

While the manuscript's overall organization is strong, there are some sentences that are challenging to understand. These issues may stem from grammatical errors, awkward phrasing, or a lack of clarity in conveying the intended message. For instance, sentences like "One of the primary functions of the LWD is to use sensors to detect the interaction between its mechanical components and the soil, where the force and deflection are induced during the operation" could be made more straightforward. Careful proofreading and editing would significantly improve the manuscript's readability.

The manuscript addresses a relevant and practical issue in geotechnical engineering by attempting to reduce the cost of evaluating soil modulus. The use of an Arduino-based signal interpretation kit to achieve this goal is innovative and aligns with the increasing trend of open-source solutions in science and engineering. However, there are areas where the manuscript could benefit from additional detail to fully convey the novelty of the approach.

I have a few open questions regarding the design:

1.       Why did you choose to use a dual microprocessor setup? The ADS1262 has an SPI (or similar) interface with a CS pin that could accommodate multiple units on a single Arduino. Alternatively, all 8 inputs on a single ADS unit could be utilized.

2.       How did you synchronize both microcontrollers? The observed error could potentially result from this lack of synchronization.

3.       You are comparing a static load on a load cell and a dynamic load. As the stress wave needs time to travel through the load cell and strain gages, the dynamic characteristics may not always match the static ones. Higher strain rates generally result in signal attenuation. Could this be the potential cause of errors? Testing the load cell's dynamic characteristics by striking it with a weight attached to an accelerometer (F = m*a) could provide insights.

4.       Can you describe the method for obtaining displacement from velocities? What are your assumptions about initial conditions?

The manuscript sometimes delves into well-known facts, which can be redundant for readers familiar with the field. It would be more advantageous to use this space to provide further context, additional examples, or practical insights, which would enhance the manuscript's value.

In summary, "A Low-Cost Lightweight Deflectometer with Arduino-Based Signal Interpretation Kit to Evaluate Soil Modulus" is an intriguing manuscript with a solid foundation. With some improvements in language and a more thorough exploration of the novelty and applications of the open-source sensor kit, this research has the potential to make a more significant contribution to the field of geotechnical engineering.

Comments on the Quality of English Language

While the manuscript's overall organization is strong, there are some sentences that are challenging to understand. These issues may stem from grammatical errors, awkward phrasing, or a lack of clarity in conveying the intended message. For instance, sentences like "One of the primary functions of the LWD is to use sensors to detect the interaction between its mechanical components and the soil, where the force and deflection are induced during the operation" could be made more straightforward. 

Author Response

COVER LETTER OF MANUSCRIPT ID Sensors-26888115

Dear Reviewers and Editors,

First of all, I would like to thank all of you for taking time to read my manuscript and sending me your insight comments and suggestions. Your insights have helped us to improve the quality of this paper. I have generally edited and checked some parts of the paper as follows:

  1. The title of the paper was changed to “A Low-Cost Lightweight Deflectometer with an Arduino-Based Signal Interpretation Kit to Evaluate Soil Modulus”
  2. I have checked all references relevant to the content of the manuscript and there are no problems.
  3. I have highlighted in yellow the revision of the manuscript due to the reviewers' comments. And I have highlighted in blue the revision of the manuscript due to the need for revision.
  4. The English writing throughout the manuscript has been extensively grammatically and semantically revised using the MDPI editing service (ID english-73852) and is certified as an attached certificate. In the debug version of the MDPI editing service, there are many small errors in the English writing that have been corrected without semantic changes, so I have not highlighted these parts.
  5. There are some revisions related to the figures and table of the paper as follows:
    • Figure 1. is completely drawn by the authors. The figure generally describes the Lightweight Deflectometer (LWD) as mentioned in ASTM E2835-11. (The citation of ASTM E2835-11 has been added in line 89 of the old manuscript and in line 90 of the revised manuscript).
    • Figure 2. is owned by the authors. The specification of products specified in references 24,25 of the old manuscript and references 21,22 of the revised manuscript.
    • Figure 4 was deleted because of its unnecessary appearance. The text is also modified due to the deletion of the figure (delete “as depicted in Figure 4” in line 192 of the old manuscript).
    • Figure 5 and Table 2 were deleted because of their unnecessary appearance. All the information can be found in Reference 33. The text is also modified due to the deletion of the figure and table (replace “depicted in Figure 5. And Table 2 shows each connection mode of the kit in detail.” by “The connection diagram and each connection mode of the kit were created following the instructions provided by Venkatesh [33].” in line 208 of the old manuscript and 203-204 of the revised manuscript).
    • Figure 11. The idea and drawing are created by the manuscript authors.
    • Figures 12, 13, and Table 4. were deleted because of their unnecessary appearance. All the information can be found in Reference 39. The text is also modified due to the deletion of the figure and table (replace “Figure 13 shows the lower and upper bounds of the particle size distribution of two typical aggregate base layers above the subbase: ASTM No. 57 and AASHTO M43 [44,45], and compares them to the particle size distribution of the measured materials.” by “The maximum particle sizes of these aggregates were 40 mm (D40) and 25 mm (D25), and they are both rhyolite rocks. D40 possesses a coefficient of uniformity of 2.88 and a coefficient of curvature of 1.19, while D25’s coefficients are 2.48 and 1.02, respectively. These materials fall within a specific gravity range of 2.67 to 2.75 [39].” In line 289 of the old manuscript and in line 284-287 of the revised manuscript).

The following are my point by point and the details of the revisions and my responses to the referees’ comments and questions.

RESPONSE TO REVIEW REPORT FORM 1

Comments and Suggestions for Authors

Your comment_1: “In abstract line 18, do not use word I, our, we and us. rephrase sentence.”

Discussion: I have changed the sentence structure and removed the word "our" in abstract line 18 (old manuscript) and in abstract line 20 (revised manuscript) without changing the meaning of the sentence as follows: “By significantly reducing costs while maintaining accuracy, our system has the potential to popularize quality control and assurance practices in the construction industry.”-> “By significantly reducing costs while maintaining accuracy, this developed system has the potential to popularize quality control and assurance practices in the construction industry.”

Your comment_2: “Line 25-28 please check how soil elastic modulus can measure by deflectometer. What is the use oedometer or triaxial test?”

Discussion: I would like to explain more detail as follows

  1. I have checked and rewritten the sentences lines 25-28 (old manuscript) and in lines 27-32 (revised manuscript) to explain more detail about how soil elastic modulus can be measured by lightweight deflectometers as follows: “Lightweight deflectometers (LWDs) have gained increasing importance in quality control and assurance testing for earthwork. They provide a rapid means of assessing the compaction quality or capacity of soils through reliable elastic modulus measurements [1-3]. One of the primary functions of the LWD is to use sensors to detect the interaction between its mechanical components and the soil, where the force and deflection are induced during the operation.” -> “The use of lightweight deflectometers (LWDs) in quality control and assurance testing for earthworks has increased in importance. They provide a rapid means of assessing the compaction quality or capacity of soils through reliable elastic modulus measurements [1-3]. One of the primary functions of LWDs is to detect the interaction between its mechanical components and the soil using sensors, where the force and deflection are induced during operation.”
  2. In this paper, we do not study and deploy the oedometer or triaxial test.

Your comment_3: “Similarly, line 39 check your statement of LWD measures compaction and bearing capacity.”

Discussion: To further clarify the statement of LWD measures compaction and bearing capacity, I would like to explain in more detail as follows:

  1. The Lightweight Deflectometer is used to quickly measure the modulus of elasticity of soils, from which compaction and bearing capacity can be assessed.
  2. The sentence has been rewritten to clarify the statement "LWD measures compaction and bearing capacity" in line 39 (old manuscript) and in lines 40-42 (revised manuscript) without changing the meaning of the sentence as follows: “LWDs have become crucial tools in construction and civil engineering due to their ability to assess soil compaction and bearing capacity quickly and accurately.”-> “Soil compaction and bearing capacity can be quickly assessed from LWD measurement results by determining the relative relationships between soil properties, making LWDs a useful tool in construction and civil engineering.”

Your comment_4: “Line 77 “This study aims at addressing the implementation and application of a cost-effective LWD sensor signal interpretation kit by leveraging open-source hardware, specifically Arduino® boards.” This is the novelty of paper then it is not suitable for Impact factor Journal. Bring it something more to make it worthy for good Journal such higher efficiency of measurement, quick etc.”

Discussion: First of all, I would like to thank you for your comments on the novelty of our paper. In order to clarify the research objectives of the paper, I have explained in more detail as follows:

  1. In this paper we use open-source hardware, specifically Arduino® boards, to create our own LWD. The most outstanding feature of open-source hardware Arduino® boards, which has made them widely used in many fields to date, is their low cost, which allows them to reach a wide range of users who do not have the resources for expensive commercial products. This is why I have included it as a new point in the paper. In addition to this feature, it also has a number of other features such as: moderate accuracy, ease of use, which were not mentioned in the manuscript.
  2. The sentence has been rewritten to add the novelty of the paper that was previously missing before in line 77 (old manuscript) and in line 79-81 (revised manuscript) as follows: “This study aims at addressing the implementation and application of a cost-effective LWD sensor signal interpretation kit by leveraging open-source hardware, specifically Arduino®”-> “This study aims to address the implementation and application of a cost-effective, moderate-precision, controllable, and regularly calibrated LWD sensor signal interpretation kit by leveraging open-source hardware, specifically Arduino® boards.”

Your comment_5: “How Authors chose critical damping (damping ratios) for the measurement? The values seem high for real structures interacted with soil.”

Discussion: The critical damping (damping ratios) introduced in the paper is geophone relative to ground velocity critical damping. Its value is provided by Geophone manufacturer (Geo Space Technologies, Inc. Houston, Tx)

Your comment_6: “There is no cross checking of results/data gathered through newly developed sensor with well-established technique. Please add”

Discussion: There is no cross checking of results/data gathered through newly developed sensor with well-established technique. I have added measurement results from literature studies on the same material that I have studied on:

  1. A result from the same well-established technique of the commercial Zorn LWD.
  2. Another result from a different method of measurement is the SSG (Soil Stiffness Gauge).

The comparative sentences have been added to the revised manuscript on lines 319-329 (revised manuscript) as follows: “The results of the newly developed LWD were systematically compared with those of Choi et al. [43], who had extensively used the Zorn LWD (ZFG 3000 GPS, Zorn Instruments [40]) and SSG tests [44] to determine the modulus of elasticity of D40 and D25 materials. Choi, Ahn, Lee, and Ahn's [43] measurements showed that for the D40 material with a dry density from 15.39 to 16.25 kN/m³, the Zorn LWD gave values between 25.50 and 42.10, while the SSG measurements ranged from 38.20 to 49.18 kN/m³. Similarly, for the D25 material with a dry density varying from 14.99 to 16.07 kN/m³, the Zorn LWD gave values between 27.41 and 41.77 and the SSG measurements ranged from 38.29 to 47.30 kN/m³. The comparative analysis showed that the results from the newly developed LWD model showed minimal deviations, falling within a narrow range when compared to the measurement techniques commonly used today.”

Comments on the Quality of English Language

Your comment: “Language of the manuscript needs improvement.”

Discussion: The manuscript was carefully revised grammatically and semantically by using MDPI English editing service.

RESPONSE TO REVIEW REPORT FORM 2

Comments and Suggestions for Authors

Your comment: “The manuscript titled "A Low-Cost Lightweight Deflectometer with Arduino-Based Signal Interpretation Kit to Evaluate Soil Modulus" presents an intriguing contribution to the field of engineering. It focuses on the development of an open-source sensor kit designed to reduce the costs associated with proprietary tools used in evaluating soil modulus. This review aims to provide an overview of the manuscript's strengths and areas that require improvement.”

Discussion: Thank you very much for your comments, suggestions, and overview of the manuscript's strengths and for pointing out the improvement part of our manuscript. Your comments helped us to see our weaknesses and to improve the paper.

Your comment: The manuscript is well-structured, providing a clear outline of the problem, methodology, results, and discussions. The introduction effectively introduces the problem statement and the significance of the research. The "Implementation of LWD" section is detailed, making it easy for readers to understand the technical aspects of the study. The results and discussions are logically organized and complemented with figures and tables, enhancing the overall readability of the manuscript. However, it's worth noting that the text in Figure 5 (schematic) is too small for reading and should be addressed.”

Discussion: Thank you very much for your comment on the structure of our paper. Regarding the issue of figure 5. I have considered its necessity and realized that it can be replaced with reference documents, so I have removed it and explained the schematic with reference documents as in the revised manuscript lines 203-205 as follows: “The connection diagram and each connection mode of the kit were created following the instructions provided by Venkatesh [33].”

Your comment: While the manuscript's overall organization is strong, there are some sentences that are challenging to understand. These issues may stem from grammatical errors, awkward phrasing, or a lack of clarity in conveying the intended message. For instance, sentences like "One of the primary functions of the LWD is to use sensors to detect the interaction between its mechanical components and the soil, where the force and deflection are induced during the operation" could be made more straightforward. Careful proofreading and editing would significantly improve the manuscript's readability.”

Discussion: Thank you very much for your comment on the overall organization and English writing of our paper. We acknowledge the shortcomings in the sentences and the English writing problems throughout the paper. We have reviewed and rewritten the sentences to make them more understandable and straightforward. For example, in lines 25-28 (old manuscript) and in lines 30-32 (revised manuscript): "One of the primary functions of the LWD is to use sensors to detect the interaction between its mechanical components and the soil where the force and deflection are induced during operation." -> " One of the primary functions of LWDs is to detect the interaction between its mechanical components and the soil using sensors, where the force and deflection are induced during operation.”

Your comment: “The manuscript addresses a relevant and practical issue in geotechnical engineering by attempting to reduce the cost of evaluating soil modulus. The use of an Arduino-based signal interpretation kit to achieve this goal is innovative and aligns with the increasing trend of open-source solutions in science and engineering. However, there are areas where the manuscript could benefit from additional detail to fully convey the novelty of the approach.”

Discussion: I would like to thank you for your comments on the novelty of our paper and to clarify the novelty of the approach, some additional details are added in line 77 (old manuscript) and in lines 79-81 (revised manuscript) as follows: “This study aims at addressing the implementation and application of a cost-effective LWD sensor signal interpretation kit by leveraging open-source hardware, specifically Arduino® boards.”-> “This study aims to address the implementation and application of a cost-effective, moderate-precision, controllable, and regularly calibrated LWD sensor signal interpretation kit by leveraging open-source hardware, specifically Arduino® boards.”

Your question_1: “Why did you choose to use a dual microprocessor setup? The ADS1262 has an SPI (or similar) interface with a CS pin that could accommodate multiple units on a single Arduino. Alternatively, all 8 inputs on a single ADS unit could be utilized.”

Discussion: To more clarify the reason of choosing a dual microprocessor setup, I would like to explain in more detail as follows:

  1. To simplify the connection of the circuits, data processing coding and to avoid collisions that could lead to errors during the measurement process of the lightweight deflectometer (LWDs).
  2. The ADS1262 is used to increase the sampling rate of the kit. I have tried to read signals from two sensors with a single Arduino and ADS1262, but the sampling rate was unstable. To stabilize the sampling rate and ensure smooth signal processing, I decided to use a dual microprocessor setup.

Your question _2: “How did you synchronize both microcontrollers? The observed error could potentially result from this lack of synchronization.”

Discussion: I would like to explain how I have synchronized the two microcontrollers and why I have neglected the possibility of a result error due to this lack of synchronization: The main operation of the LWD is to create a light impact from which its on-board sensors record the history of that impact in terms of impact force and soil velocity. The aim is to record the peak value of these two physical quantities in order to calculate the modulus of elasticity of the soil. These peaks occur almost simultaneously (the force transmitted down the plate causes deflection). The dual microprocessor is wired and set up to operate in a similar way and has the same sampling frequency (2200Hz). The dual microprocessor operates throughout the LWD measurement process, the maximum values measured by the dual microprocessor are almost simultaneous (there is a very small difference due to the difference in the time of stress transmission to the plate). For this reason, I had not considered the possibility of an error in the results due to this lack of synchronization.

Your question _3: “You are comparing a static load on a load cell and a dynamic load. As the stress wave needs time to travel through the load cell and strain gages, the dynamic characteristics may not always match the static ones. Higher strain rates generally result in signal attenuation. Could this be the potential cause of errors? Testing the load cell's dynamic characteristics by striking it with a weight attached to an accelerometer (F = m*a) could provide insights.”

Discussion: First of all, I would like to thank you for your comment on the subject of load cell calibration. It is very helpful in shedding more light on what I want to present about the situation and the process of how I calibrate the loadcell.

It is true that there is a difference in load cell calibration when the force measured is a static load on a load cell and a dynamic load. As the stress wave takes time to travel through the load cell and strain gages, the dynamic characteristics may not always match the static ones.

 The strain gauge load cell used in this paper is a highly accurate and intensive load cell, capable of capturing rapid and real-time data, which is essential for dynamic measurements. They can capture transient changes in load, making them suitable for applications where load profiles change rapidly. Where their response frequency can reach up to 5 thousand Hertz, it helps loadcell measure real-time data during the changes in applied load. In addition, during operation, when the hammer falls and collides with the buffer (conical rubber buffer), there is a delay to allow the stress wave to propagate through the buffer to the loadcell. This delay gives the loadcell enough time to settle and give accurate readings with low error. This is why I did not consider calibrating the loadcell with a dynamic load.

Your method striking loadcell with a weight attached to an accelerometer (F = m*a) is a good solution for calibrating loadcells with dynamic loads. We would like to thank you for your insightful suggestions. We have included your suggestion as an important aspect of the future plan for further research in the conclusion section. The added sentences to the revised manuscript on lines 370-373 (revised manuscript) are as follows: “In addition, several aspects of the LWD are intended to be adjusted to provide more accurate and consistent results. One of the important aspects is to enhance the dynamic measurement calibration process of the sensors, for instance using an accelerometer during loadcell calibration to account for dynamics regarding impact application and force measurement.”

Your question _4: “Can you describe the method for obtaining displacement from velocities? What are your assumptions about initial conditions?”

Discussion: I would like to explain how I obtain displacements from velocities: I have derived the displacement time history from the velocity time history to data in MATLAB, the method I have used is trapezoidal numerical integration. The input units are time with an interval of 0.00045 and the velocity time history recorded by the developed kit. The initial velocity is the velocity at t=0, v (0)=0.

Your comment: “The manuscript sometimes delves into well-known facts, which can be redundant for readers familiar with the field. It would be more advantageous to use this space to provide further context, additional examples, or practical insights, which would enhance the manuscript's value.”

Discussion: After listening to your comments. I noticed that there are some well-known facts that are generally known, so I left out some well-known facts and added some additional examples as follows:

  1. Widely use commercial ZFG 3000 GPS from Zorn Instruments.
  2. Well-known elastic modulus determination test SSG (the soil stiffness gauge).

The added sentences to the revised manuscript on lines 319-329 (revised manuscript) as follows: “The results of the newly developed LWD were systematically compared with those of Choi et al. [43], who had extensively used the Zorn LWD (ZFG 3000 GPS, Zorn Instruments [40]) and SSG tests [44] to determine the modulus of elasticity of D40 and D25 materials. Choi, Ahn, Lee, and Ahn's [43] measurements showed that for the D40 material with a dry density from 15.39 to 16.25 kN/m³, the Zorn LWD gave values between 25.50 and 42.10, while the SSG measurements ranged from 38.20 to 49.18 kN/m³. Similarly, for the D25 material with a dry density varying from 14.99 to 16.07 kN/m³, the Zorn LWD gave values between 27.41 and 41.77 and the SSG measurements ranged from 38.29 to 47.30 kN/m³. The comparative analysis showed that the results from the newly developed LWD model showed minimal deviations, falling within a narrow range when compared to the measurement techniques commonly used today.” By adding some examples from previous studies to compare with the results of the new model. I hope this will make my manuscript more valuable.

Your comment: “In summary, "A Low-Cost Lightweight Deflectometer with Arduino-Based Signal Interpretation Kit to Evaluate Soil Modulus" is an intriguing manuscript with a solid foundation. With some improvements in language and a more thorough exploration of the novelty and applications of the open-source sensor kit, this research has the potential to make a more significant contribution to the field of geotechnical engineering.”

Discussion: Thank you very much for your comments on our paper.

Comments on the Quality of English Language

Your comment: “While the manuscript's overall organization is strong, there are some sentences that are challenging to understand. These issues may stem from grammatical errors, awkward phrasing, or a lack of clarity in conveying the intended message. For instance, sentences like "One of the primary functions of the LWD is to use sensors to detect the interaction between its mechanical components and the soil, where the force and deflection are induced during the operation" could be made more straightforward.”

Discussion: The manuscript was carefully revised grammatically and semantically by using MDPI English editing service. I have edited the instance sentence that you gave me as follows: “One of the primary functions of the LWDs is to use sensors to detect the interaction between its mechanical components and the soil using sensors, where the force and deflection are induced during the operation.” In line 25-28 of the old manuscript and in line 30-32 of revised manuscript.

Your suggestions and comments have helped us a lot in the process of editing and completing this paper. Once again, we would like to thank you. We hope that you will accept our statement.

Sincerely thank you.

Round 2

Reviewer 1 Report

Comments and Suggestions for Authors

Paper has been significantly improved. All concerns of the reviewers are addressed appropriately by the authors.